# CONTINUAL LEARNING WITH RECURSIVE GRADIENT OPTIMIZATION

**Hao Liu**
Department of Computer Science
Tsinghua University
Beijing, China
`hao-liu20@mails.tsinghua.edu.cn`

**Huaping Liu**[*]
Department of Computer Science
Tsinghua University
Beijing, China
`hpliu@tsinghua.edu.cn`

## ABSTRACT

Learning multiple tasks sequentially without forgetting previous knowledge, called Continual Learning (CL), remains a long-standing challenge for neural networks. Most existing methods rely on additional network capacity or data replay. In contrast, we introduce a novel approach which we refer to as Recursive Gradient Optimization (RGO). RGO is composed of an iteratively updated optimizer that modifies the gradient to minimize forgetting without data replay and a virtual Feature Encoding Layer (FEL) that represents different network structures with only task descriptors. Experiments demonstrate that RGO has significantly better performance on popular continual classification benchmarks when compared to the baselines and achieves new state-of-the-art performance on 20-split-CIFAR100 (82.22%) and 20-split-miniImageNet (72.63%). With higher average accuracy than Single-Task Learning (STL), this method is flexible and reliable to provide continual learning capabilities for learning models that rely on gradient descent.

## 1 INTRODUCTION

In many application scenarios, one needs to learn a sequence of tasks without access to historical data, called *continual learning*. Although variants of stochastic gradient descent (SGD) have made a significant contribution to the progress made by neural networks in many fields, these optimizers require the mini-batches of data to satisfy the independent identically distributed (i.i.d.) assumption. In continual learning, the violation of this requirement leads to significant degradation of performance on previous tasks, called catastrophic forgetting. Recent works attempt to tackle this issue by modifying the training process from a variety of perspectives.

**Memory-based** approaches use extra *memory* to store some samples (Lopez-Paz & Ranzato, 2017; Chaudhry et al., 2020), gradients (Chaudhry et al., 2019a; 2020; Saha et al., 2021), or their generative models (Shin et al., 2017; Shen et al., 2020) to modify future training process. The memory for replay leads to a linear-increased space complexity with respect to the number of tasks. **Expansion-based** approaches select the network parameters dynamically (Yoon et al., 2018; Rosenbaum et al., 2018; Serra et al., 2018; Kaushik et al., 2021), add additional components as new tasks arrive (Rusu et al., 2016; Fernando et al., 2017; Alet et al., 2018; Chang et al., 2019; Li et al., 2019), or use larger networks to generate network parameters (Aljundi et al., 2017; Yoon et al., 2019; von Oswald et al., 2019). These methods reduce interference between tasks by additional task-specific parameters. Single-Task Learning (STL) can also be regarded as an expansion-based method which trains a network for each task separately. **Regularization-based** approaches encourage important parameters to lie in a close vicinity of previous solutions by introducing quadratic penalty term to the loss function (Kirkpatrick et al., 2017; Zenke et al., 2017; Yin et al., 2020) or constraining the direction of parameter update (Farajtabar et al., 2019; Chaudhry et al., 2019a; Saha et al., 2021). Our method is also regularization-based which combines the advantages of loss penalty and gradient constraint.

In this work, we focus on continual learning in a fixed-capacity network without data replay. We aim to minimize the expected increment of the total loss of past tasks without reducing the performance of the current task. To this end, we propose an upper bound of quadratic loss estimation and

---

[*]Corresponding author.

design a recursive optimization procedure to modify the direction of the gradient to the optimal solution under this upper bound. In addition, we introduce trace normalization process to guarantee the learning rate during the training process according to the principle of *current-task-first (CFT)*. This normalization process makes our approach compatible with the vast majority of existing models and learning strategies well-designed for single-task solutions. As the gradient modification process is independent of data samples and previous parameters, our optimizer can be used directly in most deep architecture networks as typical single-task optimizers like SGD. Further, to reduce the interference between tasks, we develop a feature encoding strategy to represent the multi-modal structure of the network without additional parameters. A virtual *feature encoding layer (FEL)* which randomly permutes the output feature maps using integer task descriptor as seed is attached after each real layer. Thus, each task obtains a specific virtual structure under same network parameters. Since the parameter space of the network is not changed, such a strategy will not change the fitting ability of the neural network. Experimental validations on several continual learning benchmarks show that the proposed method has significantly less forgetting and higher accuracy than existing fixed-capacity baselines. We achieve state-of-the-art performance on 20-split-CIFAR100 (82.22%) and 20-split-miniImageNet (72.62%). In addition to minimizing forgetting, this method has comparable or better performance than single-task learning which handles all tasks individually.

## 2  PRELIMINARIES

Consider $K$ sequentially arrived supervised learning tasks $\{\mathcal{T}_k | k \in [K]\}$, where $[N] := \{1, 2, \cdots, N\}$ for any positive integer $N$. In each task $\mathcal{T}_k$, there are $n_k$ data points $\{(x_{k,i}, y_{k,i}) | i \in [n_k]\}$ sampled from an unknown distribution $\mathcal{D}_k$. Let $\mathcal{X}, \mathcal{Y}, \mathcal{W}$ be the space of inputs, targets and model parameters. By denoting the predictor as $f(x, k) : \mathcal{X} \times [K] \to \mathcal{Y}$, the loss function of $\theta \in \mathcal{W}$ associated with data point $(x, y)$ and task identifier $k$ can be expressed as $l(f(\theta; x, k), y) : \mathcal{W} \to \mathbb{R}$, and the empirical loss function of task $\mathcal{T}_k$ is defined as:

$$L_k(\theta) = \frac{1}{n_k} \sum_{i=1}^{n_k} l(f(\theta; k, x_{k,i}), y_{k,i}) \tag{1}$$

In the continual learning scenario studied in this work, the parameter space $\mathcal{W}$ remains a fixed size, and an integer task descriptor is provided at both training and testing time. Without access to past samples, we use a second-order Taylor expansion to estimate the loss function of previous tasks. Let $\theta_j^*$ be the optimal parameter of $L_j(\theta)$ generated by the gradient descent process according to $\nabla_{\theta_j^*} L_j = 0$. For a new model parameter $\theta$ in the neighborhood of $\theta_j^*$, the loss of previous task $\mathcal{T}_j (j < k)$ can be estimated as:

$$L_j(\theta) = L_j(\theta_j^*) + \frac{1}{2}(\theta - \theta_j^*)^T H_j(\theta - \theta_j^*) \tag{2}$$

where $H_j := \nabla^2 L_j(\theta_j^*)$ is the Hessian matrix.

## 3  PROBLEM FORMULATION & SOLUTION

In our fixed-capacity continual learning setting, finding an appropriate joint solution that works well across the task sequence is the core goal. To this end, we introduce a novel continual learning optimization problem and corresponding iterative optimization strategy.

### 3.1  OPTIMIZATION PROBLEM

In this paper, we formalize forgetting as the the increment of old task losses. As mentioned in Section 2, the total loss of tasks before $\mathcal{T}_k$, denoted as $F_k$, can be estimated by:

$$F_k(\theta) = \sum_{j=1}^{k-1} L_j(\theta) \approx \sum_{j=1}^{k-1} [L_j(\theta_j^*) + \frac{1}{2}(\theta - \theta_j^*)^T H_j(\theta - \theta_j^*)] \tag{3}$$

As Equation (3) need the explicit value of previous model parameters, $F_k$ is too expensive to be an optimization target in continual learning. We turn to a more concise form which we refer to as *recursive least loss (RLL)*:

$$F_k^{RLL}(\theta) := \frac{1}{2}(\theta - \theta_{k-1}^*)^T (\sum_{j=1}^{k-1} H_j)(\theta - \theta_{k-1}^*) \tag{4}$$

In Appendix A.2 , we prove that $F_k^{RLL}$ and $F_k$ are equivalent for optimization if all previous tasks are fully-trained. Based on the conclusions above, the optimization problem during task $\mathcal{T}_k$ is formalized as:

$$\theta_k^*: \quad \min_\theta F_k^{RLL}(\theta), \quad \text{subject to } \nabla L_k(\theta) = 0 \tag{5}$$

$F_k^{RLL}$ has the same form as the regularization term in many regularization-based methods. The optimization goal of these methods are variants of $L_k(\theta) + \lambda F_k^{RLL}(\theta)$, derived from Bayesian posterior approximation with Gaussian prior (Kirkpatrick et al., 2017; Nguyen et al., 2018) or approximation of the KL-divergence used in natural gradient descent (Amari, 1998; Ritter et al., 2018; Tseran et al., 2018). The Bayesian methods try to estimate and minimize the overall loss function, while our method prioritizes the performance of the current task by $\nabla L_k(\theta) = 0$ and minimizes the expected forgetting of the past tasks $F_k^{RLL}(\theta)$.

## 3.2 GRADIENT MODIFICATION

For the newest task $\mathcal{T}_k$, the optimal solution where $\nabla L_k(\theta) = 0$ should be obtained by stochastic gradient descent started from the former optimal model parameter $\theta_{k-1}^*$ at the end of task $\mathcal{T}_{k-1}$. Using subscript $\cdot_i$ to represent the parameters of step $i$, and the initial state $\theta_0 = \theta_{k-1}^*$, the single step update can be expressed as:

$$\theta_i = \theta_{i-1} - \eta_i \nabla L_k(\theta_{i-1})$$

Assume that the pre-set learning rate $\eta_i$ is small enough to ignore the higher order terms, the loss function after the one-step update can be expressed as:

$$L_k(\theta_i) = L_k(\theta_{i-1}) - \eta_i(\nabla L_k(\theta_{i-1}))^T \nabla L_k(\theta_{i-1})$$

If we hope to solve the task $\mathcal{T}_k$ only, updating the parameter $\theta$ according to the gradient above is enough. However, as mentioned above, such a method will encourage the neural network to gradually forget the old tasks. Therefore, we modify the update direction to minimize the expectation of forgetting. To this end, we introduce a new positive definite symmetric matrix $P$ with appropriate dimensions to modify the gradients ($g \to Pg$). The modified one-step update is:

$$\begin{cases} \theta_i = \theta_{i-1} - \eta_i P \nabla L_k(\theta_{i-1}) \\ L_k(\theta_i) = L_k(\theta_{i-1}) - \eta_i(\nabla L_k(\theta_{i-1}))^T P \nabla L_k(\theta_{i-1}) \end{cases} \tag{6}$$

To maintain the pre-set learning rate during the continual learning problem and avoid repetitive selection of hyper-parameters, we impose an additional constraint on the trace of the projection matrix and prove the corresponding convergence rate consistency theorem 1 in Appendix A.1.

**Theorem 1** (convergence rate consistency). *Under the constraint of* trace($P$)=dim($P$), *the expectation of the learning rate for unknown isotropic distribution is the same as the original optimizer.*

For common network structures, we provide the space complexity of matrix P and the time complexity of the corresponding gradient modification process in Appendix B.2. As described above, the only extra memory of our approach is the projection matrix $P$, which contains the information of previous tasks. This allows our approach to be a space-invariant method different with typical *memory-based* or *expansion-based* continual learning method. As the performance of our method is identified with the choice of $P$, the following problem is: *How to find a good projection matrix?* We will answer this question in the following parts.

### 3.3 APPROXIMATE SOLUTION

The next step is to find a solution for problem (5). When the training process on $\mathcal{T}_k$ is finished, the final state $\theta_k^*$ and the residual loss can be obtained from the accumulation of one-step update:

$$
\begin{cases}
\theta_k^* = \theta_{k-1}^* - \sum_{i=1}^{n_k} \eta_i P \nabla L_k(\theta_{i-1}) \\
L_k(\theta_k^*) = L_k(\theta_{k-1}^*) - \sum_{i=1}^{n_k} \eta_i (\nabla L_k(\theta_{i-1}))^T P \nabla L_k(\theta_{i-1})
\end{cases}
\tag{7}
$$

In this way, for a given sample sequence and initial value $\theta_{k-1}^*$, the result of $\theta_k^*$ depends on $P$. The optimization problem 5 on $\theta_k^*$ is transformed into an optimization problem on $P$.

However, the relationship between $F_k^{RLL}(\theta_k^*)$ and $P$ is too complicated to be used in optimization process. To tackle this problem, we propose an upper bound of $F_k^{RLL}(\theta_k^*)$ as a practical optimization target.

**Theorem 2** (upper bound). *Denote $\hat{\sigma}_m(\cdot)$ as the symbol for maximum eigenvalue and $\eta_m$ as the maximum single-step learning rate, the recursive least loss has an upper bound:*

$$
F_k^{RLL}(\theta_k^*) \leq \frac{1}{2} n_k \eta_m \hat{\sigma}_m(P\bar{H}) L_k(\theta_{k-1}^*)
\tag{8}
$$

*where $\bar{H} = \sum_{j=1}^{k-1} H_j$ is defined as the sum of the Hessian matrices of all old tasks.*

We prove theorem 2 in Appendix A.3. Discarding the constant terms, we get an alternative optimization problem for projection matrix $P$:

$$
P : \begin{cases}
\min_P \quad \hat{\sigma}_m(P\bar{H}) \\
\text{subject to} \quad \text{trace}(P) = \dim(P)
\end{cases}
\tag{9}
$$

We provide a detailed solution of this problem in Appendix A.4. The normalized solution is:

$$
P = \frac{dim(\bar{H})}{trace(\bar{H}^{-1})} \bar{H}^{-1}
\tag{10}
$$

The normalized projection procedure can be described as: finding a new gradient that has similar effects on the current task to minimize the upper bound of the old task losses. Our optimizer only modifies the direction of the gradient without reducing the search region, which will guarantee the fitting ability of the network consistent throughout the task sequence.

## 4 IMPLEMENTATION

The main concern of our method in section 3 lies in the expensive space and time cost in deep neural networks. In this section, we propose two approaches to reduce the time and space complexity of the algorithm for models with forward-backward propagation structures.

### 4.1 VIRTUAL FEATURE ENCODING LAYER

In multi-layer networks, the output of the previous layer can be regarded as a set of features generated by the feature extractors(for example, weight matrices, bias vectors, convolution kernels, etc). In order to make the gradients generated in back-propagation process conform to our assumption of isotropic distribution(Theorem 1), we propose a virtual **Feature Encoding Layer(FEL)** to apply task-specific connections to the output of the previous layer and the input of the next layer.

**Definition 4.1** (Feature Encoding Layer). A feature encoding layer applies a task-specific rearrangement to the input feature maps, the order of which is randomly generated using the task identifier as a seed.

Note that FEL is only a permutation of existing feature maps, and its order does not change during the training process. Although this feature encoding layer does not require extra space, we write it in matrix form for the convenience of theoretical analysis. The permutation matrix at layer $l(l = 1, 2, \cdots, L)$ is:

$$S_l(k) := \text{random permutation of } I_l \text{ with seed}(k)$$

where $I_l$ is an identity matrix with dimensions equal to the number of feature maps. Considering that the order of the features is critical for the next layer to obtain interpretable information, the recognition ability of the network will degrade greatly without correct feature order. FEL provides an effective way to eliminate the interference between tasks, where features encoded by a specific task descriptor will be randomly permuted in other tasks. Therefore, the same feature extractor plays different roles in different tasks. Although gradients of different layer are strongly correlated in the current task, there is little correlation from the perspective of old tasks. That means current impact on past tasks can be regarded as the independent summation of the influence from different local feature extractors.

## 4.2 LOCAL-GLOBAL EQUIVALENCE

Then, during the backward propagation process of task $\mathcal{T}_k$, the gradient of $L_k$ on an intermediate layer $h_l$ can be calculated by the chain rule:

$$g_l = \frac{\partial L_k}{\partial h_l} = \frac{\partial L_k}{\partial h_L} \prod_{j=l}^{L-1} \frac{\partial h_{j+1}}{\partial h_j} = g_L \prod_{j=l}^{L-1} S_{j+1}(k) D_{j+1} W_{j+1} \tag{11}$$

where $D_{j+1}$ is a diagonal matrix representing the derivative of the nonlinear activation function. Since previous samples cannot be accessed to recalculate the gradient, common gradient-based methods (Li et al., 2019; Azizan et al., 2019; Farajtabar et al., 2019) assume that the joint optimal parameter lies in the neighborhood of the previous optimal parameter and use previously calculated gradients as an approximation, which leads to $\nabla f(\theta; x) \approx \nabla f(\theta^*; x)$ and $\frac{\partial^2 h_L}{\partial h_l^2} \approx 0$. We follow this assumption and use the proposed optimizer to ensure this assumption as much as possible. Thus we have:

$$\frac{\partial^2 L_k}{\partial h_l \partial h_r} = (\frac{\partial h_L}{\partial h_l})^T \frac{\partial^2 L_k}{(\partial h_L)^2} (\frac{\partial h_L}{\partial h_r}) \tag{12}$$

$$\bar{H}_l = \sum_{j=1}^{k-1} \frac{\partial^2 L_j}{\partial h_l^2} = \sum_{j=1}^{k-1} \sum_{i=1}^{n_j} (\frac{\partial h_L}{\partial h_l})^T l''(f(\theta; x); y) \frac{\partial h_L}{\partial h_l} + \alpha I_l \tag{13}$$

where $\alpha$ is a penalty parameter to ensure the positive definiteness of $\bar{H}_l$. We set $\alpha = 1$ in all subsequent sections according to the trace normalization proposed in Section 3.

Further, we can decompose the global optimization problem into independent sub-problems at each layer. We introduce Theorem 3 and prove it in Appendix A.5:

**Theorem 3** (Local-Global Equivalence). *Under the assumption of close vicinity, the global optimization problem is equivalent to independent local optimization problems. The local optimal projection matrix of layer $l$ is:*

$$P_l = \frac{dim(\bar{H}_l)}{trace(\bar{H}_l^{-1})} \bar{H}_l^{-1}, \quad where \ \bar{H}_l = \sum_{j=1}^{k-1} H_{j,l}$$

## 4.3 ITERATIVE UPDATE

Considering that the complexity of calculating inverse matrix with dimension of $n$ is $O(n^3)$, it is time-consuming to calculate the inverse Hessian matrix $\bar{H}_l^{-1}$ in practice. Instead, we update

the projection matrix $P_l$ iteratively like *recursive least square(RLS)* (Haykin, 2002) algorithm at training step(see Appendix B.1 for more details). This allows RGO to be an online algorithm with linear memory complexity and single-step time complexity in the number of model parameters. We summarize the gradient modification and the iterative update of the projection matrix to Algorithm 1.

Note that feature extractors in the same layer share a projection matrix calculated by the average of the gradients considering their linear correlation. In this way, handling multiple gradients in the same layer does not increase the complexity of updating projection. We list the memory size and the single step-time complexity for different kinds of feature extractors in Appendix B.2. It is worth mentioning that, because the local optimizers of different layers are independent, after obtaining the back-propagation gradients, the gradient modification process of different layers can be processed in parallel to further reduce the time required.

---

**Algorithm 1** Learning Algorithm of Recursive Gradient Optimization

---

**Input**: Task sequence $\mathcal{T}_k, k = 1, 2, \cdots$
**Output**: optimum parameter $\theta_k^*$
 1: $P_l(0) \leftarrow I_l$. $\theta \leftarrow \theta_0^*$ is randomly initialized.
 2: **for** $k = 1, 2, \cdots$ **do**
 3:     **for** $(x, y) \sim \mathcal{D}_k$ **do**
 4:         $g_l = \frac{\partial L_k}{\partial h_l}, \quad l = 1, 2, \cdots, L$             $\leftarrow$ Stochastic/Batch Gradient of current loss
 5:         $\hat{g}_l = P_l \cdot g_l \cdot \dim(P_l)/\mathrm{trace}(P_l)$             $\leftarrow$ modify origin gradients
 6:         $\theta \leftarrow \theta - \eta\hat{g}$             $\leftarrow$update the model parameters
 7:     **end for**
 8:     **for** $(x, y) \sim \mathcal{D}_k$ **do**
 9:         $g_l = [l''(f(\theta; x); y)]^{\frac{1}{2}} \frac{\partial h_L}{\partial h_l}, \quad l = 1, 2, \cdots, L$   $\leftarrow$ get local gradient by back-propagation
10:         $k_l = P_l \cdot g_l/(\alpha + g_l^T P_l g_l)$
11:         $P_l \leftarrow P_l - k_l g_l^T P_l$             $\leftarrow$Update projection matrix
12:     **end for**
13:     $\theta_k^* \leftarrow \theta$             $\leftarrow$get the optimal model parameter
14: **end for**

---

## 5 EXPERIMENT SETUP

**Benchmarks:** We evaluate the performance of our approach on four supervised continual learning benchmarks. Permuted MNIST (Goodfellow et al., 2014; Kirkpatrick et al., 2017) and Rotated MNIST (Chaudhry et al., 2020) are variants of MNIST dataset of handwritten digits (LeCun, 1998) with 20 tasks applying random permutations of the input pixels and random rotations of the original images respectively. Split-CIFAR100 (Zenke et al., 2017) is a random division of CIFAR100 into 20 tasks, each task has 5 different classes. Split miniImageNet, introduced by (Chaudhry et al., 2020), applies a similar division on a subset of the original ImageNet (Russakovsky et al., 2015) dataset.

**Baselines:** In this work, we perform experiments on the benchmarks above with the following fixed capacity methods and an expansion-based method for comparison: (1) SGD which uses stochastic gradient descent optimizing procedure to finetune the model, (2) EWC (Kirkpatrick et al., 2017) which is one of the pioneering regularization methods using fisher information diagonals as important weights, (3) A-GEM (Chaudhry et al., 2019a) which uses loss gradients of stored previous data in an in-equality constrained optimization, (4) LOS (Chaudhry et al., 2020) which constraints gradients in a low-rank orthogonal subspace, (5) ER-ring (Chaudhry et al., 2019b) which utilizes a tiny ring memory to alleviate forgetting, (6) GPM (Saha et al., 2021) which trains new tasks in the residual gradient subspace, (7) APD (Yoon et al., 2019) which is a strong expansion-based method decomposing the parameters of different tasks with a common basis, and (8) STL which trains a model for each single task. For the compared methods, we follow the original implementations to perform some necessary processing at the end of every task. The storage for memory-based methods is set to 1 sample per class per task following Chaudhry et al. (2020).

**Metrics:** We use average accuracy(ACC) and average accuracy decline, also called backward transfer(BWT) by Lopez-Paz & Ranzato (2017), to evaluate the classification performance. Denote the accuracy of task $k$ at the end of task $T$ as $R_{T,k}$, ACC and BWT are defined as:

$$\text{ACC} = \frac{1}{T}\sum_{k=1}^{T} R_{T,k}, \quad \text{BWT} = \frac{1}{T-1}\sum_{k=1}^{T} R_{T,k} - R_{k,k}$$

**Architectures and training details:** We evaluate all of the continual learning methods for the same network architectures. The model is a three-layer fully connected network with 256 hidden units in MNIST experiment and a standard ResNet18 (He et al., 2016) in CIFAR and ImageNet experiments. For RGO, we add a virtual feature encoding layer attached to each layer. MNIST variants are trained 1000 steps while CIFAR and miniImageNet are trained 2000 steps. Batchsize is set at 10 for all tasks. The task identifiers are provided for both training and testing time. All results are reported across 5 runs with different seeds. See Appendix C for more details.

## 6 RESULTS & DISCUSSIONS

Table 1: Performance of different methods on Permuted MNIST/ Rotated MNIST/ Split Cifar100/ Split mini-Imagenet. The model is trained with 20 tasks for 5 different seeds and evaluated by final average accuracy with stds. (*) denotes methods with additional network capacity. RGO-2 is a version without FEL for ablation experiments

| | | Permuted MNIST | | Rotated MNIST | |
|---|---|---|---|---|---|
| **Method** | **Replay** | $\textbf{ACC}_{test}\textbf{(\%)}$ | **BWT(%)** | $\textbf{ACC}_{test}\textbf{(\%)}$ | **BWT(%)** |
| RGO | N | **91.15**(±0.20) | -2.05(±0.09) | **91.25**(±0.01) | -1.59(±0.01) |
| RGO-2 | N | 87.95(±0.01) | -5.65(±0.38) | 72.26(±0.95) | -20.74(±0.01) |
| GPM | N | 83.29(±0.01) | -8.45(±0.01) | 70.02(±0.95) | -17.95(±0.01) |
| LOS* | N | 86.56(±0.38) | -4.10(±0.33) | 80.21(±1.11) | -13.44(±1.06) |
| ER-Ring | Y | 79.84(±0.63) | -12.88(±0.65) | 69.20(±0.79) | -25.93(±0.85) |
| AGEM | Y | 72.32(±1.04) | -19.94(±1.02) | 53.26(±1.00) | -41.74(±0.96) |
| EWC | N | 67.79(±1.60) | -24.38(±1.53) | 43.27(±0.66) | -50.74(±0.76) |
| SGD | N | 46.11(±3.91) | -46.06(±4.00) | 44.82(±0.01) | -50.18(±0.01) |
| STL | N | 91.33(±0.20) | 0.0 | 91.09(±0.01) | 0.0 |

| | | Split CIFAR100 | | Split ImageNet | |
|---|---|---|---|---|---|
| **Method** | **Replay** | $\textbf{ACC}_{test}\textbf{(\%)}$ | **BWT(%)** | $\textbf{ACC}_{test}\textbf{(\%)}$ | **BWT(%)** |
| RGO | N | **73.18**(±0.51) | -1.67 (±0.29) | **70.33**(±0.87) | -1.64 (±0.41) |
| RGO-2 | N | 62.82(±0.98) | -15.91 (±0.92) | 57.40(±1.90) | -22.40 (±1.89) |
| GPM | N | 53.41(±2.87) | -27.98 (±3.14) | - | - |
| LOS* | Y | 56.20(±1.12) | -25.46 (±1.14) | 43.25(±2.29) | -34.75 (±2.69) |
| ER-Ring | Y | 53.74(±2.13) | -28.15 (±2.02) | 45.88(±2.39) | -29.21 (±1.63) |
| AGEM | Y | 49.56(±2.64) | -32.10 (±2.73) | 34.67(±0.52) | -38.06 (±0.87) |
| EWC | N | 47.71(±1.70) | -25.17 (±1.50) | 32.61(±3.67) | -24.95 (±3.62) |
| SGD | N | 37.02(±1.64) | -44.34(±1.55) | 37.69(±1.00) | -37.23(±0.72) |
| STL | N | 74.90(±0.73) | 0.0 | 67.76(±1.70) | 0.0 |

The evolution of average accuracy is shown in Figure 1 and the final results with error bars of the indicated datasets at the end of training are reported in Table 1. The proposed method shows a strong performance of average accuracy over the baselines on all benchmarks. The result of BWT shows that RGO can significantly reduce catastrophic forgetting especially on complex tasks and deep architectures. RGO improves upon strongest baseline considerably: 17.0% and 24.5% absolute gain in average accuracy, 93.4% and 94.4% reduction in forgetting, on CIFAR100 and miniImageNet, respectively. Meanwhile, on rotated MNIST and miniImageNet, we observe that our method even exceeds the performance of STL which is often regarded as the upper bound of continual learning methods. The results of ablation experiments show that RGO maintains good performance without FEL, only slightly lower than LOS which has an additional task orthogonal mapping layer on Rotated MNIST. In Table 2, we list some results on modified LeNet from APD ((Yoon et al., 2019))

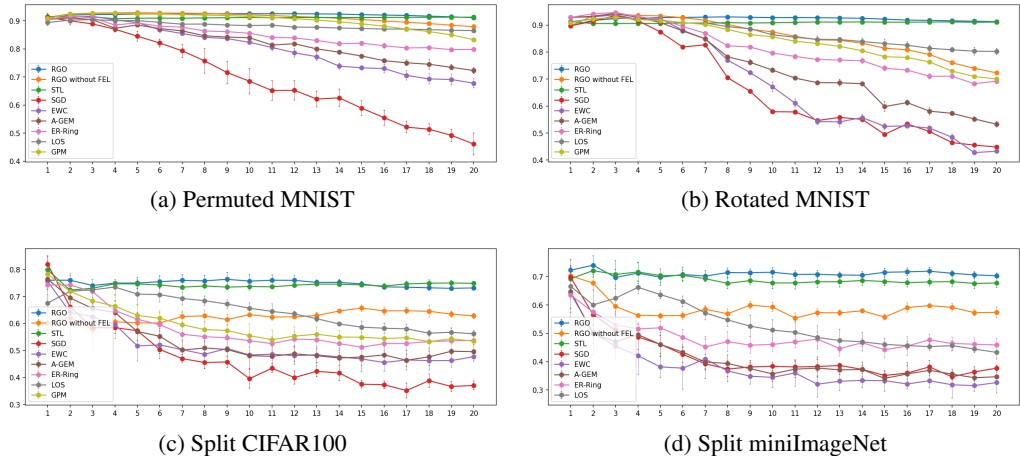

Figure 1: Evolution of average accuracy with the number of tasks during the continual learning process.

as a comparison with expansion-based methods. Contrary to the forgetting of other methods, RGO shows positive knowledge transfer and exceeds the theoretical upper bound on the testing set under a fixed network capacity.

Table 2: Average accuracy of 20-task Split-CIFAR100 dataset with modified LeNet. Capacity denotes the percentage of network capacity used with respect to the original network. Relative ACC represents the difference in accuracy between the corresponding method and STL under the same settings. (*) denotes results reported from APD.

| Metric | PGN* | DEN* | RCL* | APD* | RGO |
|---|---|---|---|---|---|
| relative ACC(%) | -10.24±0.39 | -9.90±0.77 | -9.01±0.25 | -4.19±0.33 | **+6.02±0.5** |
| Capacity | 271% | 191% | 184% | 130% | **100%** |

Further, we test our method with more architectures. As shown in Table 3, RGO achieves higher test accuracy than STL with only 5% capacity despite forgetting on the training set. RGO provides more robust features to reduce the accuracy gap between the training set and the testing set by 9% to 44%. Meanwhile, we report new state-of-the-art performance of **82.22%** and **72.63%** on Split-CIFAR100 (20 tasks) and Split-miniImageNet (20 tasks) respectively without a well-designed training schedule. Although RGO minimizes forgetting in each local optimization problem, due to the layer-by-layer accumulation of errors, deeper structures lead to more forgetting. FEL uses random permutation to greatly reduce the coupling between layers, which plays an important role in alleviating this problem. In this perspective, shallow and wide structures may be beneficial to alleviate catastrophic forgetting in the field of continual learning.

Table 3: Results of different architectures on 20-Split-CIFAR100(*) and 20-Split-miniImageNet(†). $\Delta$ denotes the difference between training set accuracy and testing set accuracy. All experiments are trained 5 times with 20 epoches. Learning rate is set at 0.03 and 0.01 for CIFAR and miniImageNet respectively.

| | Single-Task Learning | | | Recursive Gradient Optimization | | | |
|---|---|---|---|---|---|---|---|
| Architecture | $ACC_{train}(\%)$ | $ACC_{test}(\%)$ | $\Delta(\%)$ | $ACC_{train}(\%)$ | $ACC_{test}(\%)$ | $\Delta(\%)$ | BWT(%) |
| LeNet-5* | 100.0±0.00 | 75.01±0.32 | 25.0±0.3 | 99.35±0.05 | 81.03±0.51 | 18.3±0.5 | -0.99±0.15 |
| AlexNet-6* | 99.90±0.03 | 81.60±0.31 | 18.3±0.3 | 98.78±0.08 | **82.22±0.24** | 16.6±0.2 | -1.45±0.13 |
| VGG-11* | 98.30±0.55 | 79.51±0.71 | 18.8±0.6 | 95.17±0.08 | 79.81±0.22 | 15.4±0.3 | -4.00±0.19 |
| VGG-13* | 95.73±0.83 | 75.64±0.49 | 20.1±0.8 | 93.26±1.22 | 77.43±0.34 | 15.8±1.0 | -4.86±0.94 |
| AlexNet-7† | 99.72±0.30 | 71.92±0.55 | 27.8±0.6 | 98.10±0.11 | **72.63±0.45** | 25.5±0.5 | -1.98±0.16 |
| VGG-11† | 99.70±0.08 | 71.67±0.13 | 28.0±0.2 | 95.91±0.22 | 71.14±0.62 | 24.8±0.7 | -3.06±0.23 |
| VGG-13† | 97.62±0.44 | 67.22±0.76 | 30.4±0.5 | 92.48±1.10 | 66.24±1.04 | 26.2±0.7 | -4.79±0.48 |
| ResNet-18† | 97.04±0.46 | 68.78±1.09 | 28.3±0.8 | 86.79±0.85 | 71.00±0.61 | 15.8±0.3 | -5.20±0.62 |

## 7 RELATED WORK

In this section, we present some discussions between the adopted technology with existing work. The starting point of our approach and many other loss-constrained approaches is to optimize current loss and the estimated past loss at the same time. Most of the commonly used regularization methods (Kirkpatrick et al., 2017; Zenke et al., 2017; Teng et al., 2020) use a hyperparameter to balance the current task and past tasks. The objective functions of these methods can be expressed as $L(\theta) + \lambda F(\theta - \theta_{old})$. This type of method suffers from the trade-off between new tasks and old tasks and requires hyperparameter search to obtain better results. In contrast, we follow the principle of *current-task-first* discarding empirical trade-offs between tasks which means $\lambda = 0$. In the solution space of $\theta = \arg\min L$, we change the path of the gradient descent process through the P matrix and find the one that minimizes $F(\theta - \theta_{old})$. Thus, under the assumption of over-parameters, RGO optimizes current performance and forgetting simultaneously.

Although the starting point is different, our gradient modification process is closely related to gradient constraint methods like OWM (Zeng et al., 2019) and GPM (Saha et al., 2021). OWM uses similar iteratively updated projectors derived from *recursively least square(RLS)*, which regards each layer as an independent linear classifier and uses the output of the previous layer to build the projection matrix. This leads to layer-by-layer accumulation of errors in modern complex end-to-end network architectures. Using the gradients of the loss function directly, our approach is less worried about the depth of the network and more compatible with existing auto-grad frameworks. For a single-layer linear classifier $y = Wx$, OWM and RGO are equivalent considering that $\frac{\partial y}{\partial W} = x$. In addition, an extra normalization procedure in our method guarantees the learning rate of the current task. This brings an additional advantage that our method can reuse the hyperparameters of original single-task models. GPM (Saha et al., 2021) projects the gradient of each layer into a lower-dimensional residual space of previous tasks, while the parameter space of RGO is consistent for different tasks. RGO will maintain the network's fitting ability as the number of tasks increases. In addition, our method is not limited to the instability of SVD decomposition and does not require hyperparameter selection.

Task encoding layer has been used to reduce interference between tasks in recent works like LOS (Chaudhry et al., 2020) and HAT (Serra et al., 2018), which require additional network capacity. On the contrary, FEL is only a permutation of the input corresponding to the task id. This provides an efficient task encoding and decoupling method which can be easily integrated into other continual learning methods.

## 8 LIMITATION

Like all methods based on gradient constraint, analyses in this paper are based on neighborhood assumption and over-parameterized assumption which may not be satisfied in some narrower networks. When the number of tasks is close to the minimum number of channels in the network, this assumption fails. Although we have empirically proved that this effect is not obvious under common experimental settings, attention should be paid to the width of the network in applications.

## 9 CONCLUSION

In this paper, we propose a new recursive gradient optimization method to find the optimal parameters of fixed capacity networks, and a new feature encoding strategy to characterize the structure of the network. The feature encoding layer and the optimizer to minimize forgetting are both compatible with typical learning models, which allows our approach to be a general method to add continual learning capability into the vast majority of the existing network architectures learned by variants of gradient descent, with only constant times of memory/time cost than typical back-propagation algorithms. The theoretical derivation and experimental results show that RGO is currently the optimal approach under the current-task-first principle and quadratic loss estimation for fixed capacity networks. Experiments demonstrate that RGO achieves significantly better performance than other state-of-the-art methods on a variety of benchmarks. Without restrictions on the network structure and loss form, RGO has broad prospects in combination with other continuous learning methods and applications in other representation learning fields.

## REPRODUCIBILITY STATEMENT

We give the reproducible source code in the supplementary materials, and introduce the implementation of the baseline method in Appendix C.1. See Appendix C for the selection of hyperparameters. In `Python3.6` and `TensorFlow1.4`, all results can be reproduced. The theorems put forward in the main text have corresponding proofs in Appendix A.

## ACKNOWLEDGEMENT

The authors are with the Beijing National Research Center for Information Science and Technology, Institute for Artificial Intelligence, and the Department of Computer Science and Technology, Tsinghua University, Beijing, 100084, China. This work was supported by National Natural Science Fund for Key International Collaboration (62120106005).

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

## APPENDIX

Section A provides proofs of theorems introduced in the paper. Section B provides some details and approximations used in the implementation. Section C provides the details of codes, architectures, hyperparameters and resources used in the experiment.

## A PROOFS

### A.1 PROOF OF THEOREM 1

As we have mentioned, the distribution of elements of the gradients of future tasks are assumed to be isotropic, and this assumption is guaranteed by our random encoding strategy under different tasks. For isotropic distribution, we have:

**Lemma 4** (Distribution Consistency). *Isotropism is invariant under orthogonal transformation. (Larsen & Marx, 2005)*

Then, we introduce a lemma on trace of matrix which is often used in matrix analysis (Horn & Johnson, 2012):

**Lemma 5** (Trace Consistency). *Trace of matrix is consistent under orthogonal transformation.*

As described in Section 3, at $i$-th single train step during task $k$, we have:

$$\begin{cases} \theta_i = \theta_{i-1} - \eta_i P \nabla L_k(\theta_{i-1}) \\ L_k(\theta_i) = L_k(\theta_{i-1}) - \eta_i (\nabla L_k(\theta_{i-1}))^T P \nabla L_k(\theta_{i-1}) \end{cases} \tag{14}$$

For any potential gradient, the mathematical expectation of the loss function decline is:

$$\mathbb{E}[L_k(\theta_i) - L_k(\theta_{i-1})] = -\eta\mathbb{E}[(\nabla L_k)^T P \nabla L_k] \tag{15}$$

the optimizer degrades to original single task optimizer when $P$ equals to identify matrix. As positive definite symmetric matrix can be orthogonal diagonalized by a orthogonal matrix $V$,

$$VPV^T = diag\{\lambda_1, \lambda_2, \cdots, \lambda_n\} := \Lambda \tag{16}$$

where $\{\lambda_i\}$ represent the eigen values of the Projection Matrix $P$. Mark $dim(P)$ and $\nabla L_k(\theta)$ with $n$ and $x = (x_1, x_2, \cdots, x_n)$ respectively. We assume the distribution of unknown future gradients is isotropic and apply Lemma 4 to the expectation:

$$
\begin{aligned}
\mathbb{E}[(\nabla L_k)^T P \nabla L_k] &= \mathbb{E}_x[x^T V^T \Lambda_P V x] = \mathbb{E}_x[x^T \Lambda_P x] \\
&= \sum_{i=1}^{n} \lambda_i \mathbb{E}_x[x_i^2] = \sum_{i=1}^{n} \lambda_i \frac{\mathbb{E}_x[\sum_{i=1}^{n} x_i^2]}{n} \\
&= \frac{\sum_{i=1}^{n} \lambda_i}{n} \mathbb{E}_x[x^T x] = \frac{\sum_{i=1}^{n} \lambda_i}{n} \mathbb{E}[(\nabla L_k)^T \nabla L_k]
\end{aligned}
\tag{17}
$$

According to Lemma 5, the sum of the eigen values of Inverse Hessian Matrix $P$ equals to that of $\Lambda$, that means:

$$\sum_{i=1}^{n} \lambda_i = trace(P) \tag{18}$$

Thus if we add a normalize a constraint $trace(P) = dim(P)$ to the Inverse Hessian matrix $P$, every task in the continual learning procedure can have consistent expectation convergence rate.

## A.2 PROOF OF LOSS FUNCTION EQUIVALENCE

**Theorem 6** (Loss Equivalence). *If $\theta_{k-1}^* = \arg\min_\theta F_{k-1}^{RLL}(\theta)$, using $F_k^{RLL}$ or $F_k$ as the loss to optimize is equivalent, which means $\arg\min_\theta F_k^{RLL}(\theta) = \arg\min_\theta F_k(\theta)$*

We first expand $F_k$ in Equation 3 at the initial state $\theta_{k-1}^*$ during $\mathcal{T}_k$:

$$
\begin{aligned}
F_k(\theta) &\approx \sum_{j=1}^{k-1}[L_j(\theta_j^*) + \frac{1}{2}(\theta - \theta_{k-1}^* + \theta_{k-1}^* - \theta_j^*)^T H_j(\theta - \theta_{k-1}^* + \theta_{k-1}^* - \theta_j^*)] \\
&= (\theta - \theta_{k-1}^*)^T \sum_{j=1}^{k-2}[H_j(\theta_{k-1}^* - \theta_j^*)] + \frac{1}{2}(\theta - \theta_{k-1}^*)^T(\sum_{j=1}^{k-1} H_j)(\theta - \theta_{k-1}^*) + const.
\end{aligned}
\tag{19}
$$

For further analysis, we first introduce the following lemma which can be proved inductively.

**Lemma 7.** *if $(\sum_{j=1}^{k-1} H_j)(\theta_k^* - \theta_{k-1}^*) = 0$ holds for any k, then $\sum_{j=1}^{k-1}[H_j(\theta_k^* - \theta_j^*)] = 0$ also holds for any k.*

*Proof.* We mark $\sum_{j=1}^{k-1}[H_j(\theta_k^* - \theta_j^*)]$ as $D(k)$, then for $k = 1, 2, 3, \cdots$ we have

$$D(k) - D(k-1)$$

$$= \sum_{j=1}^{k-1} [H_j(\theta_k^* - \theta_j^*)] - \sum_{j=1}^{k-2} [H_j(\theta_{k-1}^* - \theta_j^*)]$$

$$= H_{k-1}(\theta_k^* - \theta_{k-1}^*) + (\sum_{j=1}^{k-2} H_j)(\theta_k^* - \theta_{k-1}^*)$$

$$= (\sum_{j=1}^{k-1} H_j)(\theta_k^* - \theta_{k-1}^*)$$

$$= 0$$

Using the fact that $D(0) = 0$, $D(k) = 0$ holds for all positive integer $k$. □

Ignoring the constant terms, the key to prove this theorem is the second term of the expression of $F_k$;

*Proof.* If $\nabla F_k^{RLL}(\theta_k^*) = 0$ satisfies for all $k \in [K]$, then we can get

$$(\sum_{j=1}^{k-1} H_j)(\theta_k^* - \theta_{k-1}^*) = 0 , \forall k \in [K]$$

Using the result of Lemma 7, we have

$$\sum_{j=1}^{k-2} [H_j(\theta_{k-1}^* - \theta_j^*)] = 0 , \forall k \in [K]$$

Substituting this formula into Equation (3) and discarding the constant terms lead to

$$F_k(\theta_k^*) = F_k^{RLL}(\theta_k^*) , \forall k \in [K]$$

Thus, $\{F_k^{RLL}\}_{k=1}^K$ and $\{F_k\}_{k=1}^K$ are equivalent loss sequences throughout the continual learning process. □

Note that this equivalence is obtained in the case of a second-order approximation of the loss functions of tasks, so it also depends on the "close vicinity" hypothesis.

### A.3 PROOF OF THEOREM 2

**Lemma 8** (Cauchy Inequality). *For any positive integer $n$, positive symmetric definite matrix $P$ and vectors $\{v_i\}_{i=1}^n$, denote the $P$-norm of $v$ by $||v||_P = \sqrt{v^T P v}$, we have:*

$$|| \sum_{i=1}^n v_i ||_P^2 \le n \sum_{i=1}^n ||v_i||_P^2$$

Denote $\hat{\sigma}_m(\cdot)$ as the symbol for finding maximum eigenvalue and $\eta_m$ as the maximum single-step learning rate, the recursive least loss has an upper bound:

$$F_k^{RLL}(\theta_k^*) = \frac{1}{2}(\sum_{i=1}^{n_k} \eta_i \nabla L_k(\theta_{i-1}))^T P(\sum_{j=1}^{k-1} H_j) P(\sum_{i=1}^{n_k} \eta_i \nabla L_k(\theta_{i-1}))$$

$$\le \hat{\sigma}_m[P(\sum_{j=1}^{k-1} H_j)]\frac{1}{2}(\sum_{i=1}^{n_k} \eta_i \nabla L_k(\theta_{i-1}))^T P(\sum_{i=1}^{n_k} \eta_i \nabla L_k(\theta_{i-1}))$$

(20)

According to Lemma 8, we have

$$F_k^{RLL}(\theta_k^*) \leq \hat{\sigma}_m[P(\sum_{j=1}^{k-1} H_j)]\frac{1}{2}n_k \sum_{i=1}^{n_k}(\eta_i \nabla L_k(\theta_{i-1}))^T P(\eta_i \nabla L_k(\theta_{i-1}))$$

$$\leq \frac{1}{2}n_k \eta_m \hat{\sigma}_m[P(\sum_{j=1}^{k-1} H_j)] \sum_{i=1}^{n_k} \eta_i(\nabla L_k(\theta_{i-1}))^T P \nabla L_k(\theta_{i-1}) \tag{21}$$

Considering that $L_k(\theta_k^*) \geq 0$ at the end of training and loss for current task before training can be expressed as $L_k(\theta_{k-1}^*)$, the change of loss function described in Equation 7 satisfies the following inequality:

$$L_k(\theta_{k-1}^*) \geq \sum_{i=1}^{n_k} \eta_i(\nabla L_k(\theta_{i-1}))^T P \nabla L_k(\theta_{i-1}) \tag{22}$$

Then we can complete the proof by combining 21 with 22.

$$F_k^{RLL}(\theta_k^*) \leq \frac{1}{2}n_k \eta_m \hat{\sigma}_m[P(\sum_{j=1}^{k-1} H_j)]L_k(\theta_{k-1}^*) \tag{23}$$

### A.4 SOLUTION OF PROBLEM 9

At a fixed point of the model, total Hessian matrix $\bar{H} := \sum_{j=1}^{k-1} H_j$ must be positive definite like $P$. As positive definite symmetric matrix can be orthogonal diagonalized by an orthogonal matrix (Horn & Johnson, 2012), $H$ and $P$ can be expressed as $\bar{H} = U^T \Lambda_{\bar{H}} U$, $P = V^T \Lambda_P V$, while $\Lambda_{\bar{H}} = \{\sigma_1, \cdots, \sigma_n\}$ and $\Lambda_P = \{\lambda_1, \cdots, \lambda_n\}$ are diagonal matrices.

Using the diagonalization step, the optimization problem 9 can be expressed as:

$$P : \begin{cases} \min_{P} & \hat{\sigma}_m(V^T \Lambda_P V U^T \Lambda_{\bar{H}} U) \\ s.t. & trace(P) = dim(P) \end{cases} \tag{24}$$

As orthogonal transformation maintains eigen value (Horn & Johnson, 2012), which means:

$$\hat{\sigma}_m(V^T \Lambda_P V U^T \Lambda_{\bar{H}} U) = \hat{\sigma}_m(U V^T \Lambda_P V U^T \Lambda_{\bar{H}})$$

There are two main variable to be optimized, diagonal matrix $\Lambda_P$ and orthogonal matrix $UV^T$. To simplify the derivation, we set $UV^T$ to the simplest orthogonal matrix $I$. Under this assumption, the optimization problem is simplified as:

$$\{\lambda_i\}_{i=1}^n : \begin{cases} \min_{\lambda_i} & \max_i \sigma_i \lambda_i \\ s.t. & \sum_{i=1}^n \lambda_i = n \end{cases} \tag{25}$$

We get the optimal eigen values:

$$\lambda_i = \frac{n}{\sigma_i \sum \frac{1}{\sigma_i}} \tag{26}$$

Thus,

$$P = V \Lambda_P V^T = \frac{n}{\sum \frac{1}{\sigma_i}} V \Lambda_{\bar{H}}^{-1} V^T = \frac{dim(\bar{H})}{trace(\bar{H}^{-1})} \bar{H}^{-1} \tag{27}$$

## A.5 Proof of Theorem 3

Denote the gradient of total parameter set and the total Hessian matrix as $g_\theta = (g_1^T, g_2^T, \cdots, g_L^T)^T$ and $\{\frac{\partial^2 L_k}{\partial \theta^2}\}_{l,r} = \frac{\partial^2 L_k}{\partial h_l \partial h_r}$, we have:

$$
\begin{aligned}
g_l^T P_l \frac{\partial^2 L_k}{\partial h_l \partial h_r} P_r g_r &= g_l^T V_l^T \Lambda_{P_l} V_l U_l^T \Lambda_{\bar{H}_l}^{\frac{1}{2}} \Lambda_{\bar{H}_r}^{\frac{1}{2}} U_r V_r^T \Lambda_{P_r} V_r g_r \\
&= g_l^T V_l^T \Lambda_{P_l}^{\frac{1}{2}} \Lambda_{P_l}^{\frac{1}{2}} \Lambda_{\bar{H}_l}^{\frac{1}{2}} \Lambda_{\bar{H}_r}^{\frac{1}{2}} \Lambda_{P_r}^{\frac{1}{2}} \Lambda_{P_r}^{\frac{1}{2}} V_r g_r \\
&\le \hat{\sigma}_m(\Lambda_{P_l}^{\frac{1}{2}} \Lambda_{\bar{H}_l}^{\frac{1}{2}}) \hat{\sigma}_m(\Lambda_{\bar{H}_r}^{\frac{1}{2}} \Lambda_{P_r}^{\frac{1}{2}}) g_l^T V_l^T \Lambda_{P_l}^{\frac{1}{2}} \cdot \Lambda_{P_r}^{\frac{1}{2}} V_r g_r
\end{aligned}
\tag{28}
$$

$$
\begin{aligned}
g_\theta^T \frac{\partial^2 L_k}{\partial \theta^2} g_\theta &= \sum_{1 \le l,r \le L} g_l^T P_l \frac{\partial^2 L_k}{\partial h_l \partial h_r} P_r g_r \\
&\le \sum_{1 \le l,r \le L} \hat{\sigma}_m(\Lambda_{P_l}^{\frac{1}{2}} \Lambda_{\bar{H}_l}^{\frac{1}{2}}) \hat{\sigma}_m(\Lambda_{\bar{H}_r}^{\frac{1}{2}} \Lambda_{P_r}^{\frac{1}{2}}) g_l^T V_l^T \Lambda_{P_l}^{\frac{1}{2}} \cdot \Lambda_{P_r}^{\frac{1}{2}} V_r g_r \\
&\le \max_l[\hat{\sigma}_m(\Lambda_{P_l} \Lambda_{\bar{H}_l})] \sum_{1 \le l,r \le L} g_l^T V_l^T \Lambda_{P_l}^{\frac{1}{2}} \cdot \Lambda_{P_r}^{\frac{1}{2}} V_r g_r \\
&= \max_l[\hat{\sigma}_m(\Lambda_{P_l} \Lambda_{\bar{H}_l})] g_\theta^T P g_\theta
\end{aligned}
\tag{29}
$$

The optimization problem above has same form as the global optimization problem in Section A.4, the solution can be easily got as:

$$
P_l = \frac{dim(\bar{H}_l)}{trace(\bar{H}_l^{-1})} \bar{H}_l^{-1}, where \quad \bar{H}_l = \sum_{j=1}^{k-1} H_{j,l}
$$

## B Implementation Details

### B.1 Details of quadratic estimation of the Hessian matrix

For the $C$-class classification problems, $f(x; \theta)$ has $C$-logits associated to different classes. We consider the most commonly used softmax cross entropy loss which is defined as

$$
l(y, f(x; \theta)) = -\sum_{j=1}^c y_j log(a_j)
\tag{30}
$$

where $a_j = exp(f_j(x; \theta)) / \sum_{c=1}^{C} exp(f_c(x; \theta))$ as the $j$-th softmax output. The $(i, j)$-th element of the second derivative matrix of the loss function with respect to $f(x; \theta)$ is then calculated as

$$
l''(y; f(x, \theta))_{i,j} = a_j \phi_{i,j} - a_i a_j
\tag{31}
$$

where $\phi_{i,j}$ is Dirac function equal to 1 while $i = j$ else 0.

As a symmetric diagonally dominant matrix, $l''(y; f(x, \theta))$ has its matrix root $[l''(y; f(x, \theta))]^{\frac{1}{2}}$. This guarantees the correctness of our algorithm. In implementation, for convenience, we only used the diagonal element corresponding to the ground truth label for an estimation.

### B.2 Time & memory complexity of RGO

We list the shape of projection matrix and time complexity of projection matrix update and gradient modification introduced by RGO for some typical feature extractors below:

First, according to Algorithm 1, the time complexity of updating $P$ is obviously O($\dim(P)^2$). The main concern comes from the matrix-matrix product in $g' = Pg$ for $g$s with higher dimension. However, if we notice the linear correlation of the columns of $g$, we can avoid this matrix multiplication. Use a fully connected layer $y = xW + b : x \in \mathbb{R}^{n_1}, y \in \mathbb{R}^{n_2}$ as an example. Considering

| Kind | Shape | Size of P | time complexity |
|---|---|---|---|
| vector | $n_1$ | $(n_1, n_1)$ | $n_1^2$ |
| matrix | $(n_1, n_2)$ | $(n_1, n_1)$ | $n_1^2$ |
| kernel | $(n_1, n_2, ksize, ksize)$ | $(ksize^2 n_1, ksize^2 n_1)$ | $ksize^4 n_1^2$ |

$\frac{\partial L}{\partial W} = \frac{\partial L}{\partial y} x^T$, we have $g' = Pg = P \frac{\partial L}{\partial y} x^T$. If we calculate from left to right instead of calculating $g$ first, we can avoid matrix multiplication and reduce the number of calculations from $n_1 n_2 + n_1^2 n_2$ to $n_1 n_2 + n_1^2$. The amount of calculation beyond the original backpropagation is only $n_1^2$. The calculation process for kernels is the same except for a reshape process.

Considering that both the kernel size and $\frac{n_1}{n_2}$ have upper bounds in common neural network models, the time complexity of RGO remains the same as that of backpropagation.

## C  EXPERIMENT DETAILS

### C.1  BASELINE IMPLEMENTATIONS

EWC (Kirkpatrick et al., 2017), LOS (Chaudhry et al., 2020), A-GEM (Chaudhry et al., 2019a), and ER-ring (Chaudhry et al., 2019b) are implemented from adapting the code provided by Chaudhry et al. (2020) under MIT License. GPM is implemented from the official implementation provided by Saha et al. (2021) under MIT License.

### C.2  RESOURCES

All experiments of our method are completed in several hours with 4 pieces of Nvidia-2080Ti GPUs.

### C.3  ARCHITECTURES

We provide details of architectures we used in the experiment section.

- LeNet-5: A modified LeNet used by Yoon et al. (2019). There are two convolutional layers with kernels size of (5,5) and channels of (20,50), followed by two hidden fully connected layer with (800,500) units.
- AlexNet-6: A modified AlexNet used by Saha et al. (2021). There are three convolutional layers with kernels size of (4,3,2) and channels of (64,128,256), followed by two hidden fully connected layer with (2048,2048) units.
- AlexNet-7: A modified AlexNet. There are four convolutional layers with kernels size of (5,4,3,3) and channels of (64,128,128,128), followed by two hidden fully connected layer with (2048,2048) units.
- VGG-11&VGG-13 : Original VGG11 and VGG13 proposed by Simonyan & Zisserman (2015).
- ResNet-18: A standard 18-layer ResNet proposed by He et al. (2016). For our approach, we remove all batch-norm layers because their parameters are not updated by gradient descent.

LeNet-like and AlexNet-like architectures are attached a 2×2 maxpooling layer after each convolutional layer.

### C.4  HYPERPARAMETERS

The learning rates of all baselines are generated by hyperparameter search in [0.003,0.01,0.03,0.1,0.3,1] to achieve better results. Other hyperparameters of EWC, A-GEM, ER-Ring and LOS follows Chaudhry et al. (2020), while those of GPM and APD follows their official implementation.

- Single Task Learning
    - learningrate: 0.1(MNIST), 0.03(CIFAR100, miniImageNet)

- Recursive Gradient Optimization(Ours)
    - learningrate: 0.1(MNIST), 0.03(CIFAR100, miniImageNet 2000steps), 0.01(miniImageNet 20epochs)
- SGD
    - learningrate: 0.1(MNIST), 0.03(CIFAR100, miniImageNet)
- EWC
    - learningrate: 0.1(MNIST), 0.03(CIFAR100, miniImageNet)
    - regularization: 10(MNIST, CIFAR100, miniImageNet)
- A-GEM
    - learningrate: 0.1(MNIST), 0.03(CIFAR100, miniImageNet)
- ER-Ring
    - learningrate: 0.1(MNIST), 0.03(CIFAR100, miniImageNet)
- LOS
    - learningrate: 0.1(MNIST), 0.4(CIFAR100), 0.2(miniImageNet)
- GPM
    - learningrate: 0.1(MNIST), 0.03(CIFAR100, miniImageNet)
    - threshold: 0.95 for first layer and 0.99 for other layers(MNIST) ,increase from 0.97 to 1(CIFAR), increase from 0.985 to 1(miniImageNet)
    - dimension of representation matrices: 300(MNIST), 125(CIFAR), 100(miniImageNet)

