# OpenReview forum: "Continual Learning with Recursive Gradient Optimization"
_ICLR.cc/2022/Conference — ICLR 2022 Spotlight_

### Official Review · Reviewer_Ztre · 2021-10-31

**Correctness:** 4
**Technical Novelty And Significance:** 4
**Empirical Novelty And Significance:** 4
**Recommendation:** 8
**Confidence:** 4

**Main Review:**

This paper provides an empirically very strong method for task-based continual learning, as measured by average task accuracy, alongside a solid theoretical derivation and justification for it, which will be of interest to the continual learning community and for these reasons I am recommending it for acceptance. The main weakness of the method is that it has a runtime that is quadratic in the number of parameters at each layer, as a result of the gradient projection at each update; the paper could greatly benefit from an empirical analysis of the runtime compared to competing methods in order to quantify this limitation.

Positives:
- Very strong empirical performance on standard CL benchmarks, exhibiting essentially no forgetting across 20 tasks.
- The method is simple to implement.
- Ensuring that the expected learning rate on the current task is preserved (via setting trace(P)=dim(P) and with the feature encoding layers) reduces the need for hyperparameter tuning, in contrast to other regularization methods that often require tuning of hyperparameters that control the balance of progress on the current task and preservation of previous task performance.
- Despite the quadratic runtime in the number of parameters per layer, the projections of the gradients at each layer are independent of each other and so can be computed in parallel (once the gradients have been calculated via backprop).
- While the method requires knowledge of task ids for the random permutation layers, these only require storage of a single integer seed each. It would be interesting to investigate how this method could be extended to a task-free setting.
- Good discussion of related methods

Potential areas for improvement / questions:
- The runtime of the algorithm is quadratic in the number of parameters per layer, which could be concerning given that it is acknowledged in the paper that the method requires wide enough network layers to satisfy the overparameterisation assumption. A runtime comparison across methods should be conducted in order to quantify the differences.
- It’s a bit difficult to understand what the method does from the introduction. It could do with a diagram.
- In Table 1, shouldn’t the values in the BWT be negative as in Table 3?

Typos:
- Equation 4 typo: $\theta_{k-1^*}$ -> $\theta_{k-1}^*$
- Section 4 title: “Implemention” -> “Implementation”
- 4.1 Line 2 “matrixs” -> “matrices”
- Definition 4.1 “applys” -> “applies”


**Summary Of The Paper:**

This paper introduces a gradient-based approach to continual learning in neural networks called Recursive Gradient Optimization (RGO). RGO modifies the gradient direction at each update by multiplying it with a projection matrix P that is designed to minimize the increase in loss on previously encountered tasks. RGO is theoretically designed to prioritise performance on the current task and, among the optimal solutions for the current task, find the one that causes least interference with previous tasks - thus it assumes that the network is overparameterised. The derivation of the method starts by approximating the continual learning loss with the “recursive least loss”, which involves a sum of the Hessians of the previous tasks and assumes that tasks are fully trained and their solutions are close by to each other. It is then shown that the recursive least loss can be upper-bounded by an expression involving the projection matrix P, via which the solution for P that minimizes this upper bound is derived. It is shown that the expectation of the step size can be preserved by guaranteeing that trace(P)=dim(P) and by introducing random task-specific permutations at each layer - this preserves the “current-task-first” principle and reduces the need for hyperparameter tuning. Experiments are run on a number of standard continual learning image classification benchmarks, demonstrating an extremely strong performance of RGO in comparison to competing methods, sometimes surpassing the performance of a baseline that trains a separate model for each task.

**Summary Of The Review:**

This paper provides an empirically very strong method for task-based continual learning, as measured by average task accuracy, alongside a solid theoretical derivation and justification for it, which will be of interest to the continual learning community and for these reasons I am recommending it for acceptance. The main weakness of the method is that it has a runtime that is quadratic in the number of parameters at each layer, as a result of the gradient projection at each update; the paper could greatly benefit from an empirical analysis of the runtime compared to competing methods in order to quantify this limitation.

---

> ### Author Response · Authors · 2021-11-11
> **Reply on time complexity**
>
>
> We are sorry that the description of time complexity in the previous text is not detailed enough, but only mentions "time complexity is the same as backpropagation". There is a technique that is easy to overlook but we have not emphasized. Without it, the time complexity of the algorithm will be overestimated. Thanks for your reminder. We have analyzed the time complexity in more detail in the Appendix B.2 of the latest version, hoping to alleviate your worries about the runtime.
>
> The main concern about runtime comes from the matrix-matrix product in $g'=Pg$ for $g$s with higher dimension. However, if we notice the linear correlation of the columns of $g$, we can avoid this matrix multiplication. Use a fully connected layer $y= xW+b: x\in \mathbb{R}^{n_1}, y\in \mathbb{R}^{n_2}$ as an example. Considering $\frac{\partial L}{\partial W} = \frac{\partial L}{\partial y} x^T$, we have $g' = Pg = P\frac{\partial L}{\partial y} x^T$. If we calculate from left to right instead of calculating $g$ first, we can avoid matrix multiplication and reduce the number of calculations from $n_1n_2+n_1^2n_2$ to $n_1n_2+n_1^2$. The amount of calculation beyond the original backpropagation is only $n_1^2$. The calculation process for kernels is the same except for a reshape process.
>
> Considering that both the kernel size and $\frac{n_1}{n_2}$ have upper bounds in common neural network models, the time complexity of RGO remains the same as that of backpropagation.

---

> > ### Comment · Reviewer_Ztre · 2021-11-18
> > **Reply on time complexity**
> >
> > Thank you for your reply, this makes more sense now. It might be useful to have an empirical runtime comparison in any case, but the theoretical clarification here helps.

---

### Official Review · Reviewer_DhW7 · 2021-11-02

**Correctness:** 3
**Technical Novelty And Significance:** 3
**Empirical Novelty And Significance:** 2
**Recommendation:** 5
**Confidence:** 4

**Main Review:**

- I agree with the validity of the Taylor expansion in equation (2). However, I find the assumption in Eq(3), that \theta is in the vicinity of each of \theta_j^*, doubtful. This is as if we say that the optimal parameters for each task are in the vicinity of each other which would abolish the problem of catastrophic forgetting from the beginning.
- The approach seems to depend on two main new components (1) The new projected matrix P, and (2) the FEL layer. I am still unsure which of the two reduces forgetting. The authors are encouraged to define two new baselines each of which incorporates one of the aforementioned components.
- There is a discrepancy between Algorithm 1 and the previously derived quantities. Concretely, the square root in line 9, and I am also missing the inverse of the sum of Hessians, H_l, in Algorithm 1.
- Is there a missing sum in the first equation of (14)?


**Summary Of The Paper:**

The paper proposes a continual learning approach based on recursive gradient optimization. To this end, a projection matrix is derived for the gradient modification. This matrix, P, is computed incrementally and updated by integrating the Hessian on each task locally.


**Summary Of The Review:**

I have two issues with (i) the assumptions made in parts of the derivations, and (ii) the missing important baselines that should be derived from your approach.

---

> ### Author Response · Authors · 2021-11-11
> **Reply to the comments.**
>
> Thanks for your review!
>
> 1.
> Because in our setting, we cannot access the past data to obtain the gradient at the current parameter, so we can only assume that the gradient obtained at the past parameter is still meaningful after the parameter changes. Under this setting, all gradient-based methods need this assumption. As you said, the goal of staying in the neighborhood is the same as reducing forgetting. To train new tasks will inevitably lead to the destruction of the neighborhood hypothesis. How to reduce this conflict is the research goal of all continual learning methods. RGO believes that moving in different directions will bring different degrees of damage, and RGO tries to find the best direction. There is no definitive indicator to indicate when the neighborhood hypothesis will be destroyed (or catastrophic forgetting will happen) unless all samples are accessed all the time, otherwise continual learning will no longer be a problem. This is similar to some control problems, such as using a linear PID model to control an inverted pendulum. Of course in some extreme cases, the set control method will fail, but it is far more difficult to study when it will fail than to design the controller itself. The upper bound of forgetting that we proposed is important progress in this regard. It may be one of the solutions for the problem that all continual learning methods need to face. Minimizing this upper bound is a good guarantee for the neighborhood hypothesis.
>
> 2.
> We have actually given the results of using FEL and not using FEL to study how much additional improvement FEL brings. The results are shown in Table 1. Even if FEL is not used, RGO exceeds all baselines, except for the results of LOS in Rotated MNIST. But LOS includes an additional trainable layer to map different tasks, which is the same as FEL. Therefore, this layer should be removed for fairness when compared with RGO without FEL. Hope this partly solves your doubts.
>
> FEL will reduce the layer-by-layer accumulation of forgetting in deeper networks and reduce the correlation between tasks. Including the commonly used setting of a separate classification layer for each task, FEL is also a similar technique and does not require additional space. At the same time, the proposal of FEL in this article is the natural requirement of RGO to deal with each layer as independent sub-problems. Although this is a general method that can be easily combined with other methods, FEL is actually part of RGO. Since RGO is equal to RLS algorithm under a single-layer linear model, we believe that RGO is also the most suitable method for applying FEL. As we mentioned, the two components of this article can be easily combined with other algorithms and have almost no negative effects. So the application of RGO in other algorithms is also an interesting research direction, which will be the future research direction but this is not the focus of this paper.
>
> 3. P_l is the inverse of the sum of Hessians and the matrix inversion process has been replaced by the RLS algorithm. The square root is obtained by symmetrically decomposing the update amount in Eq. 14, considering $g_l^Tg_l = (\frac{\partial h_L}{\partial h_l})^T l''(f(\theta;x);y) \frac{\partial h_L}{\partial h_l}$.
>
> 4. Thanks! We have revised the typo you find out in Eq. 14.

---

### Official Review · Reviewer_CQxg · 2021-11-03

**Correctness:** 4
**Technical Novelty And Significance:** 4
**Empirical Novelty And Significance:** 3
**Recommendation:** 8
**Confidence:** 4

**Main Review:**

Paper strengths:
* Continual learning is an important, challenging and practical problem which the authors have addressed. The problem is formulated as tackling forgetting (increment of old task losses) with the constraint of learning optimally for the new tasks.
* The theoretical analysis of the problem (forgetting as increment of old task losses, its approximation to more tractable RLL and further to its upperbound) is done very good.
* FEL with task specific random permutation of features is interesting and shown to be effective by decorrelating the features between different tasks but with strong correlation inside individual tasks.
* Being a theoretically motivated work, the authors have done a very good job in clearly motivating the different steps with derivations and explanations as and when needed.
* The experiments show the superiority of the approach very good.

Paper weaknesses:
* This is regarding eqn. (4). I think I got the major steps of this derivation. It uses the definition of $F_{k-1}(\theta^*_{k-1})$ and the fact that Hessian is symmetric. However, I wonder if there would be an extra $L_{k-1}(\theta^*_{k-1})$ term at the end. It would be helpful if a more detailed derivation is provided for.
* Though appendix B.2 provides space complexity of projection matrix, I am interested on the advantage of RLL formulation in terms of the capacity of the network. $F_k$ in (4) and $F^{RLL}_k$ in (5) both need to store all $H_j$, I see. Any comment specific to network capacity in relation to this would be good to have.
* Minor typos: First line of eqn. (4) one of the $\theta$s have the asterisk as subscript. Page 4 last paragraph and section 5 first paragraph – ‘applys’ should ‘applies’.

**Summary Of The Paper:**

The paper addresses a long-standing problem of deep neural networks where new training data for new tasks arrive continuously and data does not stay forever (continual learning). The network has to learn to cater for all the tasks without forgetting what it learnt for an older task as data from older tasks are no longer available. The authors came up with a formulation where learning will mean minimizing the (expectation of) forgetting and forgetting is formalized as increase of losses corresponding to the old tasks. The authors then move on to find a Taylor series expression of forgetting and an equivalent Recursive Least Loss (RLL) formulation of the loss (aka forgetting). Next the authors introduce a modification in the direction of gradient update of normal SGD that makes sure to minimize forgetting. The modification implied the introduction of a positive definite matrix and the optimization problem transformed into an optimization problem on finding the optimal positive definite matrix minimizing the RLL. As the relation between RLL and the positive definite matrix is not straightforward, an upper bound of RLL is proposed and the optimal solution is found under this upper bound. One more interesting proposal is a virtual feature encoding layer (FEL) that applies task-specific random rearrangement to the input feature maps. FEL eliminates possible interference between tasks by decorrelating features among old tasks without expanding the memory footprint. The experiments show great improvement over sota approaches on benchmark datasets and sometimes (rotated MNIST, Split ImageNet) even goes beyond supposed upperbound performance given by single task learning showing evidence of positive transfer.  Overall, I liked the approach, the presentation and the experimental results with only a few minor concerns detailed below.

**Summary Of The Review:**

In summary, the paper presents a crisp solution to a challenging and important problem with mostly clear writing and experiments. Some of the concepts introduced here (e.g., gradient direction change and FEL) can be useful not only in continual learning research but applications where forgetting can be an issue or decorrelating features can be advantageous.

---

> ### Author Response · Authors · 2021-11-11
> **Reply to the comments.**
>
> Thanks for your kind review!
> 1. We did miss a constant term, which remains unchanged during the training process of subsequent tasks, so it does not affect the correctness of the algorithm. Revisions have been made in the new version.
> 2.  Although we used the sum of $H$ in the formula derivation process, we did not use a single $H$ in any place, so what needs to be stored is the sum of the $H$s of the past tasks instead of all of them. Therefore, the memory required by RGO does not increase with the number of tasks. Moreover, in Algorithm 1, this sum is also unnecessary, the only thing need to be stored is the inverse matrix P.
> 3. Thanks for the typos!

---

### Official Review · Reviewer_oYYz · 2021-11-03

**Correctness:** 3
**Technical Novelty And Significance:** 3
**Empirical Novelty And Significance:** 3
**Recommendation:** 8
**Confidence:** 4

**Main Review:**

Overall, I think this is a nice idea and a good application of it. It is relevant for ICLR, and the idea is novel (specifically, as far as I am aware, no one has previously used results such as Theorem 1 and Feature Encoding Layers). It is nice to be able to modify the gradient on a new task with a projection matrix, while minimising forgetting on past tasks. The experiments seem comprehensive in range (many benchmarks, many architectures).

In general, the paper introduces the algorithm well and the order of information seems nice. I focus the rest of the review on various ways I think the paper could be improved further.

1. The authors argue that "Single-Task Learning ... can be regarded as the strongest expansion-based method" (Introduction), and compare a lot to STL (in the abstract and experiments). I disagree that STL is the strongest baseline, as there is no potential for forward or backward transfer between tasks with STL. In fact, many algorithms can beat STL for this exact reason (STL has even sometimes been considered as a 'lower bound', see for example [1] and the Split CIFAR task they have, which is slightly different to the one here). Overall, I believe STL is a strong method, but it is not quite right to say it is the "strongest", and perhaps the authors should not emphasise comparing so much to it throughout the paper.

2. Throughout the paper, there are missing references to many Bayesian-inspired approaches to continual learning. Examples include EWC [2], online EWC [3], VCL [4,5], Online Structured Laplace [6], FROMP [1], and many many more. These are usually regularisation-based approaches ([1] also uses memory). They are based off Bayes equation for continual learning. A key benefit is that Equations 4 and 5 naturally "drop out" of applying Bayes equation (with the Laplace approximation), and I am not sure the proofs (such as Appendix A.2) are required -- people should already see that Equation 5 holds from Bayes (in particular, online EWC / online structured Laplace use the same equation). A Bayesian view also justifies the use of alpha in Section 4.2, where alpha is the prior precision (or weight-decay parameter when training the 1st task). There is of course a key difference between the Bayesian-inspired approaches and this work: the Bayesian approaches minimise the overall loss together L_k + F_k^RLL, whereas this work minimises F_k^RLL subject to \grad L_k = 0. The authors can make this clear when discussing the Bayesian approaches.

3. Modifying the direction of gradients is closely related to natural-gradient descent (Amari, 1998) and mirror-descent. See for example NCL [7] where they discuss some of this.

4. I believe a term is missing from Equation 4 (bottom line): $L_{k-1}(\theta_{k-1}^*)$

5. The authors re-use hyperparameters from previous work for baselines. Unfortunately baselines such as EWC are known to be sensitive to hyperparameter values (for EWC, this is the regularisation parameter lambda) when the benchmark and/or architecture changes. It therefore seems unfair to report results based off one value. It would be good if the authors ran a hyperparameter sweep for the baselines too for fairness, and reported in the text that additional computational effort was needed. I still expect RGO to significantly outperform baselines.

6. In Appendix B.1, what do the authors mean by "we only used the diagonal element corresponding to the ground truth label for estimation". Why did the authors not use Equation 31 itself? I thought Equation 31 was cheap to calculate and use (requiring a forward pass only through the model).

7. [Minor point] It was interesting for me to see that "deeper structures lead to more forgetting" (page 8). I would love to see further investigations / intuition as to why this is the case with RGO (or maybe in general). Perhaps this can be future work.

Typos:
- Section 2: the authors first introduce W as the space of model parameters, but for the rest of the paper use only theta.
- Section 4.1 page 4: "matrixs" -> "matrices", "applys" -> "applies"
- Algorithm 1 line 9: missing "l=1,2,...,L"
- Table 2: please say the full names of all the metrics/methods reported in this table, along with references to the papers. Perhaps this information can go in the caption.



References

[1] Pan et al., 2020, "Continual deep learning by functional regularisation of memorable past"
[2] Kirkpatrick et al., 2017
[3] Schwarz et al., 2018, "Progress & Compress..."
[4] Nguyen et al., 2018, "Variational Continual Learning"
[5] Loo et al., 2021, "Generalized Variational Continual Learning"
[6] Ritter et al., "Online structured Laplace approximations..."
[7] Kao et al., 2021, "Natural continual learning..."

**Summary Of The Paper:**

The paper introduces a new method, Recursive Gradient Optimization (RGO), for continual learning in the task-incremental scenario. This method modifies the direction of gradients on a new task in order to minimise forgetting on previous tasks, and unlike many previous works, does not require storing past raw data to do so. The authors introduce a Feature Encoding Layer to achieve this. The authors provide experiments on 4 benchmarks of varying size (MNIST to miniImageNet), showing good performance of their algorithm, with different architectures.

**Summary Of The Review:**

Overall, I am leaning towards accept as I like the idea and the results are very strong. I have provided potential ways to improve the paper in my main review, specifically points 2 and 5.

*****************
After rebuttal: I have increased my score to 8, thanks to the authors rephrasing certain parts / adding certain comments/references to the paper. I believe the paper could still be increased by tuning hyperparameters of baselines, however, this does not detract from the quality of the proposed method.

---

> ### Author Response · Authors · 2021-11-11
> **Reply to the comments.**
>
>
> Thank you very much for your comments, most of which are valuable for this paper and our future work. In response to your concerns, we give the following replies:
>
> 1. Is STL the upper bound of expansion-based methods?
>
> The requirement of the expansion-based method mentioned in the article is that the number of sample visits is consistent with the STL, otherwise it will be classified as a memory-based method. The article you cited [1] stores 200 samples for each task in Split Cifar10, which is a memory-based method rather than an expansion-based method. In addition, according to Algorithm 1, this method needs to sequentially iterate the sample subsets of all past tasks in each update of the current task. The time complexity is quadratic of the number of tasks, which is not strictly "continual". Although we have listed some memory-based methods for comparison, the amount of calculation of these methods is linear with the number of tasks. Therefore, we believe that this method's good performance is due to its much more frequent access to old samples. In addition, the algorithm of this paper does not regard STL as a lower bound as you said, it shows a significant drop when the number of saved samples decreases according to Figure 3(c). In summary, we still believe that there is currently no method to exceed STL under the same number of sample visits, except for our method in some cases.
>
> 2. Can RGO be derived from a Bayesian perspective?
>
> Similar to your idea, we have also made a lot of effort to try to derive RGO from the extreme conditions of Bayesian viewpoint (such as setting the weight of the current task to infinity). But there are two key conflicts. (a) Enlarging the weight of the current task will aggravate the forgetting of the past task. (b) In Equation 5, it is inevitable that the earlier task parameters will appear instead of only $\theta_{k-1}$. Therefore, the gap with Bayesian method is not so easy to cross, although it looks similar.
>
> 3.
>
> About the derivation from equation (3) to equation (6), if there are only two tasks, using the method similar to NGD or the method proposed will be the same. The reason for the extra derivation in this part of this paper is to make the derivation from two tasks to multitasking more natural (or require weaker assumptions).
>
> 4. The missing $L_{k-1}(\theta_{k-1}^*)$
>
> We did miss a constant term, which remains unchanged during the training process of subsequent tasks, so it does not affect the correctness of the algorithm. Revisions have been made in the new version.
>
> 5. Hyperparameter reuse
>
> As you said, the hyperparameters we used are the results of the previous paper [2], and the experimental settings in this article are slightly different from those in the previous article. The difference is that we used 20 tasks instead of 17 tasks, the other settings are the same. If we only focus on the results of the first 17 tasks, we can reuse the optimal hyperparameters searched out completely, and Figure 1. shows that there is no difference in the relative results. The training strategy of GPM is more complicated, including a well-designed parameter schedule, which makes it difficult to search for hyperparameters. For fairness, in the parameter selection of RGO, we also did not perform hyperparameter search but directly used the most commonly used learning rates of other algorithms. We believe that insensitivity to hyperparameters is also an advantage of RGO, and algorithms that require careful adjustment of hyperparameters for tasks are not reliable enough. RGO has outperformed other baselines by a large margin, and we believe that different hyperparameters will not affect the relative results. The hyperparameter search for all experiments at the same time requires high computational resources and takes a long time. We have performed a single hyperparameter search on the MNIST dataset, and the optimal value has not changed. I hope we have answered your doubts. If you still think that we need to perform hyperparameter searches on all experiments, I will try my best to get all the results before the final version.
>
> 6. Why not use Equation 31 itself?
>
> The code in the attachment is implemented using TensorFlow with a static graph structure. The process of finding matrix roots is difficult to be added to the automatic derivation framework. If the process of finding matrix roots is omitted, it will be much easier to implement and the performance is good enough.
>
> 7.
> Temporarily, we believe that the reason is the amplification caused by the multiplication of the multi-layer parameter changes. This is indeed a direction worthy of further exploration.

---

> > ### Comment · Reviewer_oYYz · 2021-11-19
> > **Reviewer response**
> >
> > Thanks to the authors for the clarifications / discussion.
> >
> > 1. I think it is fair to say that, theoretically speaking, STL is not an upper-bound for expansion-based approaches. This is because STL cannot use forward and backward transfer. Consider the simple example of training on MNIST (with 60,000 inputs and 10 classes). Say the continual learning benchmark here involves 6000 tasks, each task with 10 images each, one image from each class. STL would clearly perform badly here, as it is not able to use forward and backward transfer. If the authors are looking for empirical results on larger scales, then reference [5] ("Generalized VCL") also has the same Split-CIFAR benchmark, and shows expansion-based approaches (HAT and GVCL-F) performing better than STL. See Figure 4(b) in [5].
> >
> > 2. I agree that RGO cannot be derived from a Bayesian perspective. As I said in my review, Bayesian approaches optimise a different loss, and do not have a constraint like in RGO. However, the proof that the authors use for Equations 4 and 5 is remarkable similar to the reasoning that Bayesian approaches use (with a Laplace approximation). I think the paper would be stronger (and easier understood by many in the community) by linking to such previous works. Note that a Bayesian approach strictly only involves $\theta_{k-1}$, and *does not* involve other previous task parameters. Please see VCL [4] and Online EWC [3]. The original EWC paper [1] does have previous task parameters, but it is not correct to call that Bayesian.
> >
> > 3. I think the paper would be stronger by having a couple of lines referencing NGD and mirror-descent approaches. I do not think the authors need to re-write their section, they should just mention related work so the reader can see some intuition.
> >
> > 4. Thanks.
> >
> > 5. I understand the reasoning to not tune hyperparameters for the baselines, it makes sense. However, some readers will question why algorithms they have used before are performing worse than expected. I mentioned tuning some key hyperparameters as it is known that some of these algorithms do not perform well unless these hyperparameters are tuned. This is indeed a key limitation in the current algorithms. I leave it to the authors to decide whether or not it is worth the (admittedly annoying) effort to do this.
> >
> > 6. Thanks. I think some previous works have used Equation 31 but assumed that the matrix is constant, ie do not backpropagate through the square root. So, just use the matrix instead of the true labels, still keeping it constant (or maybe just use the diagonal terms from the matrix). This is just a suggestion that the authors might want to consider trying in the future, as it might lead to improved results.

---

> > > ### Author Response · Authors · 2021-11-22
> > > **Authors response**
> > >
> > > We have deleted the claim about "STL is upper bound of expansion based method" and added some discussion about Bayesian method and natural gradient descent (See Section 3.1). Thanks for your kind review!

---

> > > > ### Comment · Reviewer_oYYz · 2021-11-22
> > > > **Reviewer response**
> > > >
> > > > Thanks.
> > > >
> > > > A minor note here: in Section 3.1, I don't agree that 'the results of these methods will not be optimal for the current task'. This is because I believe that forward (/backward) transfer is crucial to an *optimal* continual learning algorithm, even though most of the current methods are struggling with avoiding catastrophic forgetting. If there was no forward/backward transfer, then optimising just for the current task is optimal. If there is transfer, then, according to a Bayesian, Bayes rule is optimal (of course, the highly restrictive approximations made prevent this). Optimising just for the current task loss (even with constraints), as you do in this paper, is not necessarily optimal. In summary: I would avoid using the word "optimal", which is quite strong, and which I disagree with.

---

> > > > > ### Author Response · Authors · 2021-11-22
> > > > > **A more appropriate description**
> > > > >
> > > > > Thanks!
> > > > > We replaced it by a more appropriate description "The Bayesian methods try to estimate and minimize the overall loss function, while our method prioritizes the performance of the current task by $\nabla L_{k}({\theta}) = 0$ and minimizes the expected forgetting of the past tasks $F_k^{RLL}(\theta)$."

---

### Public Comment · ~Shengyang_Sun1 · 2021-11-19
**Interesting Paper**

This is an interesting paper!

A quick question, for one layer with $n_i$ input neurons and $n_o$ output neurons, what is the shape of the preconditioning matrix $P$ and $\hat{g}_l$? Is $P$ of size $(n_i n_o) \times (n_i n_o)$ or $n_o \times n_o$ ? It seems to be the latter based on L5 in Alg 1, but how $\hat{g}_l$ is then used to the update the weight matrix $W_l$ ?

---

> ### Author Response · Authors · 2021-11-22
> **Authors response**
>
> Thanks for your interest in our work! The size is $n_0\times n_0$. Please see Appendix B.2 for more details on the update process and corresponding space&time complexity.

---

### Public Comment · ~Cantona_ViVian1 · 2022-01-31
**Interesting Paper**

Thank for the paper.

I have one question:

1) "The storage for memory-based methods is set to 1 sample per class per task".

This is too restrictive for memory-based methods. With such a setting, I doubt memory-based methods would be more memory-efficient. In Table 1, AGEM has a MUCH lower score compared with the original paper.  How about increasing the number of sample per class for memory-based methods?

---

> ### Public Comment · ~Hao_Liu18 · 2022-02-20
> **Response on memory-efficiency**
>
> The increase of memory does improve the performance of memory-based methods. Taking CIFAR100 as an example, the results reported in the original AGEM article are higher than that reported in this paper (49.6% -> 62.3%), but still much lower than the results of RGO (~ 73% on ResNet18, up to 82.22% on AlexNet). The performance of AGEM's original paper mainly lies in: 1. It uses 65 times the memory space. 2. There are three tasks for pre-training. 3. There are three fewer tasks for the continual learning test.
>
> On the problem of memory efficiency, we want to point out that AGEM depends on the size of the data sample, the number of samples stored in each task and the number of tasks, while the memory space of RGO only depends on the model, and the space does not increase with the number of tasks. It is also one of the advantages of RGO. So we only did experiments on a variety of different models in this paper.
>
> Based on this difference, all three parts of the space need to be considered when comparing the memory-efficiency two methods. Taking CIFAR100 as an example, RGO with AlexNet-6 is the most memory efficient and most accurate result. Compared with the original AGEM results on reduced ResNet18, the model is simpler, the additional memory size is much smaller, and the average accuracy is ~20% higher.
>
> We recommend choosing an appropriate model when using RGO. Smaller models may not only reduce storage space, but sometimes improve accuracy.

---

### Public Comment · ~Yavuz_Faruk_Bakman1 · 2023-01-22
**Interesting Paper**

Thanks for your work. However, I'm confused about the notation in the paper. When you say 'layer', do you mean parameters of the model in the layer or activations in the layer? in Equation 11, it seems, you mean activations of layer and g_l corresponds to gradient for the activations of the layer because this is the only way we can apply such a chain rule. However, in the rest of the paper g_l corresponds to the gradient for the parameters in layer l. May you make it more clear for me? Thanks

---

### Decision · Program_Chairs · 2022-01-20

**Decision:**

Accept (Spotlight)

**Comment:**

This paper proposes an innovative method for continual learning that modifies the direction of gradients on a new task to minimise forgetting on previous tasks without data replay. The method is mathematically rigorous with a strong theoretical analysis and excellent empirical results across multiple continual learning benchmarks. It is a clear accept. There was good discussion between the reviewers and authors that addressed a number of minor issues, including clarifying that the method has the same computational complexity as backpropagation. The authors are encouraged to make sure that these points are addressed in the final version of the paper.